# The Flow Matrix Offers a Straightforward Alternative to the Problematic Markov Matrix

Jessica Strzempko [1,2] and Robert Gilmore Pontius, Jr. [1,*]

1   School of Geography, Clark University, 950 Main Street, Worcester, MA 01610, USA;
    jessstrzempko@gmail.com
2   Tetra Tech, 159 Bank Street, Third Floor, Burlington, VT 05401, USA
*   Correspondence: rpontius@clarku.edu; Tel.: +1-508-793-7761

**Abstract:** The Flow matrix is a novel method to describe and extrapolate transitions among categories. The Flow matrix extrapolates a constant transition size per unit of time on a time continuum with a maximum of one incident per observation during the extrapolation. The Flow matrix extrapolates linearly until the persistence of a category shrinks to zero. The Flow matrix has concepts and mathematics that are more straightforward than the Markov matrix. However, many scientists apply the Markov matrix by default because popular software packages offer no alternative to the Markov matrix, despite the conceptual and mathematical challenges that the Markov matrix poses. The Markov matrix extrapolates a constant transition proportion per time interval during whole-number multiples of the duration of the calibration time interval. The Markov extrapolation allows at most one incident per observation during each time interval but allows repeated incidents per observation through sequential time intervals. Many Markov extrapolations approach a steady state asymptotically through time as each category size approaches a constant. We use case studies concerning land change to illustrate the characteristics of the Flow and Markov matrices. The Flow and Markov extrapolations both deviate from the reference data during a validation time interval, implying there is no reason to prefer one matrix to the other in terms of correspondence with the processes that we analyzed. The two matrices differ substantially in terms of their underlying concepts and mathematical behaviors. Scientists should consider the ease of use and interpretation for each matrix when extrapolating transitions among categories.

**Keywords:** category; extrapolation; land change; flow; model; Markov

## 1. Introduction

Scientists want to extrapolate dynamic systems where portions of the extent transition from one category to another category. The Markov matrix offers an approach that assumes a constant proportion of a losing category transitions to each other category during all time intervals [1]. The Markov matrix is popular in several land change models [2], such as the CLUE family of models [3–5], DINAMICA [6–9], Markov-FLUS [10], the MOLUSCE plugin for QGIS [11,12], and TerrSet's Land Change Modeler and Cellular Automata—Markov model [13–19]. Extensions of the Markov matrix to land change modeling systems include Markov Chain Random Fields [20,21].

Scientists routinely rely exclusively on the Markov matrix despite the method's substantial mathematical challenges. These include the difficulty of extrapolation to desired time points [22,23]. Nevertheless, the Markov matrix is entrenched in software and the minds of modelers perhaps due to ignorance of an alternative. Our article presents the Flow matrix, which offers an alternative to the Markov matrix as a method to describe and extrapolate transitions among categories. The Flow matrix extrapolates on the time continuum by assuming each categorical transition has a constant size per increment of continuous time [24].

Our manuscript compares the Flow matrix with the Markov matrix in terms of mathematical properties and implications for applications. We demonstrate the characteristics of each approach using a simple example and also data from wetland and suburban landscapes in Massachusetts, USA. The two applications relate to land change while the concepts are not limited to land change.

## 2. Materials and Methods

### 2.1. Illustrative Example

Figure 1 gives example data for the Raw matrix and the resulting Flow and Markov matrices. The Raw matrix gives the size $c_{ij}$ that transitions from the start category $i$ in each row to the end category $j$ in each column during a calibration time interval that starts at time $t_0$ and ends at time $t_1$. The Sum column in the Raw matrix gives each category's start size. Each entry in the Flow matrix is the size of transition per unit of time. The Flow matrix does not have diagonal entries because diagonal entries indicate persistence, not change. The results for the Flow matrix in Figure 1 assume the duration of the calibration time interval is a single unit of time. The Loss column on the right gives the loss per time unit for each category, while the Gain row on the bottom gives the gain per time unit for each category. The entry at the bottom right in the Flow matrix gives the total change per time unit by summing all entries and subtracting the diagonal entries of the Raw matrix, where the number of categories is $J$. Each Markov matrix entry is the corresponding Raw matrix entry divided by the category's start size, regardless of the calibration interval's duration. The Markov proportions in each row have the same denominator, so the sum of the Markov entries in each row is one.

| (a) Raw | Category 1 | Category 2 | Category 3 | Sum |
|---|---|---|---|---|
| Category 1 | $c_{11} = 32$ | $c_{12} = 8$ | $c_{13} = 0$ | $s_1(0) = 40$ |
| Category 2 | $c_{21} = 0$ | $c_{22} = 16$ | $c_{23} = 4$ | $s_2(0) = 20$ |
| Category 3 | $c_{31} = 8$ | $c_{32} = 0$ | $c_{33} = 32$ | $s_3(0) = 40$ |

| (b) Flow | Category 1 | Category 2 | Category 3 | Loss |
|---|---|---|---|---|
| Category 1 | | $\dfrac{c_{12}}{t_1 - t_0} = 8$ | $\dfrac{c_{13}}{t_1 - t_0} = 0$ | $\dfrac{c_{12} + c_{13}}{t_1 - t_0} = 8$ |
| Category 2 | $\dfrac{c_{21}}{t_1 - t_0} = 0$ | | $\dfrac{c_{23}}{t_1 - t_0} = 4$ | $\dfrac{c_{21} + c_{23}}{t_1 - t_0} = 4$ |
| Category 3 | $\dfrac{c_{31}}{t_1 - t_0} = 8$ | $\dfrac{c_{32}}{t_1 - t_0} = 0$ | | $\dfrac{c_{31} + c_{32}}{t_1 - t_0} = 8$ |
| Gain | $\dfrac{c_{21} + c_{31}}{t_1 - t_0} = 8$ | $\dfrac{c_{12} + c_{32}}{t_1 - t_0} = 8$ | $\dfrac{c_{13} + c_{23}}{t_1 - t_0} = 4$ | $\dfrac{\left\{\sum_{i=1}^{J} \sum_{j=1}^{J} c_{ij}\right\} - \sum_{j=1}^{J} c_{jj}}{t_1 - t_0} = 20$ |

| (c) Markov | Category 1 | Category 2 | Category 3 | Sum |
|---|---|---|---|---|
| Category 1 | $\dfrac{c_{11}}{\sum_{j=1}^{3} c_{1j}} = .8$ | $\dfrac{c_{12}}{\sum_{j=1}^{3} c_{1j}} = .2$ | $\dfrac{c_{13}}{\sum_{j=1}^{3} c_{1j}} = 0$ | 1 |
| Category 2 | $\dfrac{c_{21}}{\sum_{j=1}^{3} c_{2j}} = 0$ | $\dfrac{c_{22}}{\sum_{j=1}^{3} c_{2j}} = .8$ | $\dfrac{c_{23}}{\sum_{j=1}^{3} c_{2j}} = .2$ | 1 |
| Category 3 | $\dfrac{c_{31}}{\sum_{j=1}^{3} c_{3j}} = .2$ | $\dfrac{c_{32}}{\sum_{j=1}^{3} c_{3j}} = 0$ | $\dfrac{c_{33}}{\sum_{j=1}^{3} c_{3j}} = .8$ | 1 |

**Figure 1.** The example data in the format of the (**a**) Raw matrix, (**b**) Flow matrix, and (**c**) Markov matrix. The gray cells are the matrix entries, and the white cells explain the entries. The rows of each matrix are the categories at the calibration interval's start time $t_0$, and the columns are categories at the end time $t_1$. Numerical values assume the duration of the calibration time interval is one.

Figure 2 uses the example data to illustrate the differences between the Flow extrapolation in the left column of graphs and the Markov extrapolation in the right column of graphs. The vertical axis shows the size as a percentage of the extent. The horizontal axis shows the time from the start of the calibration interval, which is time 0. The end of the calibration interval is time 1. The extrapolation is from time 1 to time 5. The Flow matrix portrays change through continuous time. The Markov matrix portrays change through incremental time intervals and thus shows category sizes at distinct time points. Readers will find it helpful to visualize the concepts in Figure 2 before seeing the details of the mathematics in Section 2.2.

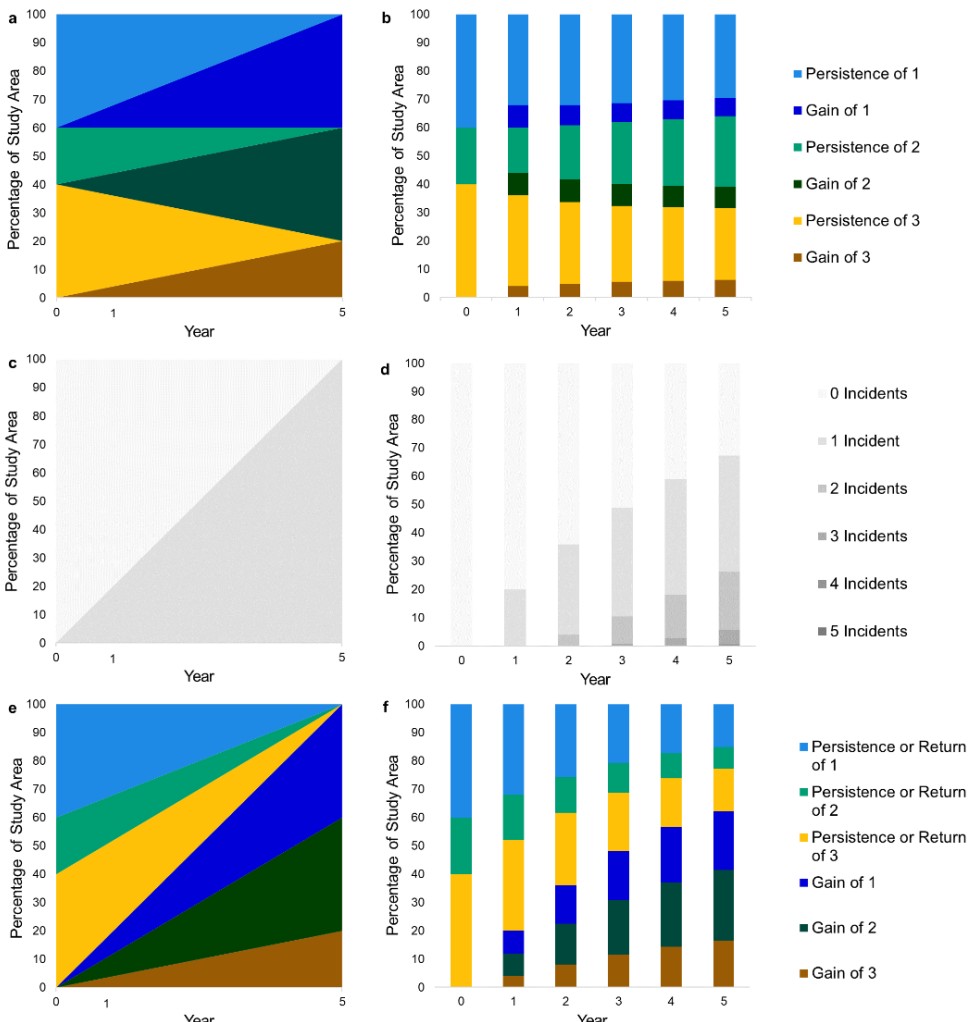

**Figure 2.** The three graphs on the left show the Flow extrapolation while the three graphs on the right show the Markov extrapolation from a calibration interval that starts at time 0 and ends at time 1. One legend applies to each pair of graphs. The upper pair shows the size of each category via (**a**) Flow and (**b**) Markov, where Markov shows persistence and gain from the preceding time point. The middle pair shows the cumulative number of incidents via (**c**) Flow and (**d**) Markov. The lower pair shows the temporal difference from time 0 in the bottom three segments via (**e**) Flow and (**f**) Markov.

We design Figure 2a,b to show the size of each category at each time point by giving Category 1 at the top, stacked above Category 2, stacked above Category 3 at the bottom. Figure 2a,b begin with the size of the categories at time 0. A category's size at a time point derives from two components: persistence and gain. Shrinkage of a category's persistence implies its gross loss. A Flow extrapolation ends when a category's persistence shrinks

to zero, which occurs at year 5. The Markov extrapolation in Figure 2b shows the sizes at each time point in terms of the persistence and gain during the time interval from the preceding time point, not from time 0. The Flow extrapolation shows that category 1 maintains its size, category 2 grows, while category 3 shrinks. The Markov extrapolation shows that Categories 1 and 3 shrink while Category 2 grows as each category approaches a constant size.

Figure 2c,d show the accumulation of incidents from time zero as a percentage of the extent. An incident occurs when a portion of the extent experiences change during a time interval. All portions of the extent have either zero or one incident for Flow extrapolation. Figure 2c shows that the entire extent accumulates one incident at time 5 for the Flow extrapolation. In contrast, the Markov extrapolation allows portions of the extent to change repeatedly during the sequence of time intervals. This occurs when a category gains and then subsequently loses, which occurs for all three categories in the example data of Figure 1. When this occurs, the Markov extrapolation accumulates an additional incident during each time interval; thus, the maximum number of accumulated incidents at the end of each time interval is the number of time intervals. Figure 2d shows that the extent for the Markov extrapolation accumulates up to 5 incidents at time 5.

We design Figure 2e,f to show the temporal difference from time 0. The gain segments are at the bottom of the graphs, so the reader can visualize the overall temporal difference as the sum of the bottom three segments. The Flow extrapolation shows a linear increase of temporal difference while the Markov extrapolation shows a deceleration of temporal difference as time progresses. For the Flow extrapolation, Figure 2a,e contain identical segments but in a different sequence of the vertical stack. For the Markov extrapolation, the gain segments in Figure 2b are smaller than the gain segments in Figure 2f because Figure 2b shows the change from the previous time point whereas Figure 2f shows the temporal difference from the start of the calibration interval. The legend's label of "Gain" denotes a portion of the study area where the category at the time point is different from the category at time 0. The legend's label of "Persistence or Return" denotes a portion of the study area where the category at the time point is the same as the category at time 0. Persistence is the portion of a category that never changes from time 0. Return is the portion of a category at time 0 that loses and then subsequently returns to the original category. Return can occur with a Markov extrapolation but not with a Flow extrapolation because Return requires more than one incident.

Section 2.2 gives the equations that express the concepts mathematically. The equations in Section 2.2 produce figures in the format of Figure 2. Readers should refer to Figures 1 and 2 while reading Section 2.2.

### 2.2. Equations

2.2.1. Raw Matrix

Table 1 gives the notation that the equations use. Italics indicate a variable; bold indicates a matrix; and round parentheses indicate a function. In the equations, square brackets indicate a matrix while curly braces indicate the order of operations.

Equation (1) defines Matrix $\mathbf{C}$ as the raw matrix for the calibration time interval starting at time $t_0$ and ending at time $t_1$. Matrix $\mathbf{C}$ contains $c_{ij}$ to denote the size of the transition from category $i$ at time $t_0$ to category $j$ at time $t_1$. We assume that at least one off-diagonal value in $\mathbf{C}$ is positive. Figure 1a illustrates $\mathbf{C}$ and an additional column on the right that gives the size of category $i$ at time $t_0$. The size of category $j$ at time $t_1$ is the sum of the entries in column category $j$ of $\mathbf{C}$.

$$\mathbf{C} = \begin{bmatrix} c_{11} & \cdots & c_{1J} \\ \vdots & \ddots & \vdots \\ c_{J1} & \cdots & c_{JJ} \end{bmatrix} \tag{1}$$

**Table 1.** Mathematical notation.

| Variable | Description |
| --- | --- |
| $c_{ij}$ | Size of transition from category $i$ at time $t_0$ to category $j$ at time $t_1$ |
| **C** | Raw matrix with $J$ rows, $J$ columns, and entries $c_{ij}$ |
| **D**$(n)_d$ | Family of row vectors for time interval $n$ where each member of the family corresponds to a specific $d$ with $d = 0, 1, \ldots, n$. Each row vector has $J$ columns where each entry is a category's size at $t_n$ that derives from $d$ incidents from $t_0$ to $t_n$. |
| $d$ | Number of incidents which is an integer on the interval $[0, n]$ defined as the number of times a pixel's category has transitions from time $t_0$ to $t_n$ |
| $f_{ij}$ | Size per unit of time for the transition from category $i$ at time $t_0$ to a different category $j$ at time $t_1$ |
| **F** | Flow matrix with $J$ rows, $J$ columns, and entries $f_{ij}$ |
| **G** | Gain matrix with $J$ rows, $J$ columns, and entries $m_{ij}$ off the diagonal and zeroes on the diagonal |
| **H**$(n)$ | Matrix with $J$ rows, $J$ columns, and the same entries as $\mathbf{M}^n$ off the diagonal and zeroes on the diagonal |
| $i$ | Index for a category at the start time of a time interval |
| **I** | Identity matrix, which has 1 for each diagonal entry and 0 elsewhere |
| $j$ | Index for a category at the end time of a time interval |
| $J$ | Number of categories >1 |
| $m_{ij}$ | Proportion of category $i$ at time $t_0$ that transitions to category $j$ at time $t_1$ |
| **M** | Markov matrix with $J$ rows, $J$ columns, and entries $m_{ij}$ |
| $n$ | Index for the time interval from time $t_{n-1}$ to time $t_n$ where $n = 1, 2, \ldots \infty$ |
| **P** | Persistence matrix with $J$ rows, $J$ columns, and entries $m_{ii}$ on the diagonal and zeroes off the diagonal |
| **R**$(n)$ | Matrix with $J$ rows, $J$ columns, and the same entries as $\mathbf{M}^n$ on the diagonal and zeroes off the diagonal |
| $s_i(0)$ | Size of category $i$ at time $t_0$ |
| $s_i(n)$ | Size of category $i$ at time $t_n$ |
| **S**$(0)$ | Row vector with $J$ columns and entries $s_i(0)$ |
| **S**$(n)$ | Row vector with $J$ columns and entries $s_i(n)$ |
| $t$ | Continuous time on the interval $[t_0, T]$ |
| $t_0$ | Start time of the calibration time interval |
| $t_1$ | End time of the calibration time interval |
| $t_n$ | End time of time interval $n$, which is also the start time of interval $n + 1$ |
| $T_i$ | Time when category $i$ reaches zero persistence for the Flow extrapolation |
| $T$ | Time when the Flow extrapolation stops |
| $w_{ij}(t)$ | Size of transition from category $i$ at time $t_0$ to category $j$ at time $t$ |
| $w_{jj}(t)$ | Size of persistence of category $j$ from time $t_0$ to time $t$ |
| **W**$(t)$ | Matrix with $J$ rows, $J$ columns, and entries $w_{ij}(t)$ for the Flow extrapolation |

### 2.2.2. Flow Matrix

Equations (2)–(6) describe the Flow extrapolation. If category $i$ is different from category $j$, then Equation (2) computes $f_{ij}$ as $c_{ij}$ divided by the duration of the calibration time interval. Equation (3) defines the Flow matrix **F**, which collects the $f_{ij}$ in the off-diagonal entries. **F** does not have diagonal entries. Equation (4) gives entries $w_{ij}(t)$ which are the sizes of transition from category $i$ at time $t_0$ to category $j$ at time $t$ according to the Flow extrapolation. If category $i$ is different than category $j$, then $w_{ij}(t)$ is the product of $f_{ij}$ and the duration from time $t_0$ to time $t$. If category $i$ is the same as category $j$, then $w_{ii}(t)$ is the size of the persistence of category $i$ from time $t_0$ to time $t$. Variable $w_{ij}(t)$ is the size of transition in the same units as the raw matrix **C**. The Flow extrapolation applies from time $t_0$ to time $T$. The time $T$ when the Flow extrapolation stops is the earliest time when a category reaches zero persistence. Equation (5) computes $T_i$ as the time when category

$i$ reaches zero persistence. We derive Equation (5) by setting $w_{ii}(t)$ to zero in Equation (4) and then solving for $t$. Equation (6) gives $T$ as the minimum value of $T_i$ over all categories.

$$f_{ij} = \left\langle \begin{array}{c} \text{if } i \neq j \text{ then } f_{ij} = c_{ij}/\{t_1 - t_0\} \\ \text{else } f_{ij} \text{ does not exist} \end{array} \right. \tag{2}$$

$$\mathbf{F} = \begin{bmatrix} & \cdots & f_{1J} \\ \vdots & \ddots & \vdots \\ f_{J1} & \cdots & \end{bmatrix} \tag{3}$$

$$w_{ij}(t) = \left\langle \begin{array}{c} \text{if } i \neq j \text{ then } w_{ij}(t) = f_{ij}\{t - t_0\} \\ \text{else } w_{ii}(t) = s_i(0) - \frac{\{s_i(0) - c_{ii}\}\{t - t_0\}}{\{t_1 - t_0\}} \end{array} \right. \tag{4}$$

$$T_i = \left\langle \begin{array}{c} \text{if } s_i(0) \neq c_{ii} \text{ then } T_i = t_0 + \frac{\{t_1 - t_0\}s_i(0)}{s_i(0) - c_{ii}} \\ \text{else } T_i = \infty \end{array} \right. \tag{5}$$

$$T = \text{MINIMUM}(T_i) \text{ over all } i \tag{6}$$

Equation (7) defines matrix $\mathbf{W}(t)$ which organizes entries $w_{ij}(t)$. Equation (8) gives the Flow gain of category $j$ from time $t_0$ to time $t$ computed as the sum down column $j$ of Matrix $\mathbf{W}(t)$ minus the diagonal entry $w_{jj}(t)$. Incidents are the number of times a portion of the extent transitions through time. Figure 2a,e each have $2J$ segments. The $J$ persistence segments are the $J$ diagonal entries of $\mathbf{W}(t)$. The $J$ gain segments are the $J$ results from Equation (8). Figure 2c derives from Equations (9) and (10), which sum to the size of the spatial extent. Equation (9) gives the size of zero incidents in the extent as the summation of each category's persistence from time $t_0$ to time $t$. Equation (10) gives the size of one incident in the extent as the sum of the entries in matrix $\mathbf{W}(t)$ minus the sum of the diagonal entries of $\mathbf{W}(t)$.

$$\mathbf{W}(t) = \begin{bmatrix} w_{11}(t) & \cdots & w_{1J}(t) \\ \vdots & \ddots & \vdots \\ w_{J1}(t) & \cdots & w_{JJ}(t) \end{bmatrix} \tag{7}$$

$$\text{Flow gain of } j \text{ from time } t_0 \text{ to time } t = \left\{ \sum_{i=1}^{J} w_{ij}(t) \right\} - w_{jj}(t) \tag{8}$$

$$\text{Size of zero incidents for Flow from time } t_0 \text{ to time } t = \sum_{j=1}^{J} w_{jj}(t) \tag{9}$$

$$\text{Size of one incident for Flow from time } t_0 \text{ to time } t = \sum_{j=1}^{J} \left\{ \left\{ \sum_{i=1}^{J} w_{ij}(t) \right\} - w_{jj}(t) \right\} \tag{10}$$

### 2.2.3. Markov Matrix

Equations (11)–(17) describe the Markov extrapolation. Equation (11) gives $s_i(0)$ as the size of category $i$ at time $t_0$ by summing across row $i$ of matrix $\mathbf{C}$. The results from Equation (11) produce the row vector $\mathbf{S}(0)$ in Equation (12), which gives the sizes of each category at the start of the calibration interval. If the size of category $i$ at time $t_0$ is not zero, then Equation (13) computes $m_{ij}$ as the proportion of category $i$ at time $t_0$ that transitions to category $j$ at time $t_1$. If the size of category $i$ at time $t_0$ is zero and $i \neq j$, then Equation (13) computes $m_{ij}$ as zero, which portrays a situation where category $i$ never transitions to a different category. Equation (14) gives the Persistence Matrix $\mathbf{P}$ where the diagonal entries are $m_{ii}$ and the off-diagonal entries are zero. Equation (15) gives the Gain Matrix $\mathbf{G}$ where the off-diagonal entries are $m_{ij}$ and the diagonal entries are zero. Equation (16) gives the Markov matrix $\mathbf{M}$ as the sum of matrix $\mathbf{P}$ and matrix $\mathbf{G}$. The row vector $\mathbf{S}(n-1)$ times the Persistence matrix $\mathbf{P}$ produces a row vector that gives the size of persistence from time $t_{n-1}$

to time $t_n$ for each category. The row vector $\mathbf{S}(n{-}1)$ times the Gain Matrix $\mathbf{G}$ produces a row vector that gives the size of gain from time $t_{n-1}$ to time $t_n$ for each category. The Markov extrapolation is performed iteratively across multiple time intervals where the duration of each time interval equals the duration of the calibration time interval. The end time point of a Markov iteration is denoted as time $t_n$ where $n$ is the index for the time interval from time $t_{n-1}$ to time $t_n$ where $n = 1, 2, \ldots \infty$. Variable $n$ is the number of iterations in the Markov extrapolation. The size of category $i$ at time $t_n$ is $\mathbf{S}(0)$ times the Markov matrix $\mathbf{M}$ raised to the power $n$. Equation (17) gives $\mathbf{S}(n)$ which is a row vector of category sizes at time $t_n$. The vector is the row vector $\mathbf{S}(n{-}1)$ times Markov matrix $\mathbf{M}$. The $J$ persistence segments in Figure 2b are the $J$ entries of the row vector that derives from the product $\mathbf{S}(n{-}1)\mathbf{P}$. The $J$ Gain segments in Figure 2b are the $J$ entries of the row vector that derive from the product $\mathbf{S}(n{-}1)\mathbf{G}$.

$$s_i(0) = \sum_{j=1}^{J} c_{ij} \tag{11}$$

$$\mathbf{S}(0) = \begin{bmatrix} s_1(0) & \cdots & s_J(0) \end{bmatrix} \tag{12}$$

$$m_{ij} = \left\langle \begin{array}{l} \text{if } s_i(0) \neq 0 \text{ then } m_{ij} = c_{ij}/s_i(0) \\ \text{else if } i \neq j \text{ then } m_{ij} = 0 \text{ else } m_{ii} = 1 \end{array} \right. \tag{13}$$

$$\mathbf{P} = \begin{bmatrix} m_{11} & \cdots & 0 \\ \vdots & \ddots & \vdots \\ 0 & \cdots & m_{JJ} \end{bmatrix} \tag{14}$$

$$\mathbf{G} = \begin{bmatrix} 0 & \cdots & m_{1J} \\ \vdots & \ddots & \vdots \\ m_{J1} & \cdots & 0 \end{bmatrix} \tag{15}$$

$$\mathbf{M} = \begin{bmatrix} m_{11} & \cdots & m_{1J} \\ \vdots & \ddots & \vdots \\ m_{J1} & \cdots & m_{JJ} \end{bmatrix} = \mathbf{P} + \mathbf{G} \tag{16}$$

$$\mathbf{S}(n) = \mathbf{S}(n-1)\mathbf{P} + \mathbf{S}(n-1)\mathbf{G} = \mathbf{S}(n-1)\mathbf{M} \tag{17}$$

Figure 2d derives from an iterative algorithm that Equation (18) describes. Equation (18) gives row vector $\mathbf{D}(n)_d$ for which the entries are the size of each category that has $d$ incidents after $n$ time intervals. The number of incidents $d$ is the number of times a portion of the extent changes category from time $t_0$ to time $t_n$. Variable $d$ is a whole number bounded on the inclusive interval from zero to $n$. The top part of Equation (18) shows that the vector of zero incidents from the previous time point $t_{n-1}$ multiplied by the Persistence matrix $\mathbf{P}$ gives the category sizes that have zero incidents at a time point $t_n$. All observations have zero incidents at time $t_0$; therefore, we define row vector $\mathbf{D}(0)_0$ as the row vector $\mathbf{S}(0)$. The middle part of Equation (18) requires the sum of persistence and gain during the previous time interval. The size of each category to persist during time interval $n$ with $d$ incidents is the product of the incident vector at time $t_{n-1}$ with $d$ incidents and the Persistence matrix $\mathbf{P}$. The size of each category that transitions during time interval $n$ and increases from $d{-}1$ to $d$ incidents is the product of the incident vector at time $t_{n-1}$ with $d{-}1$ incidents and the Gain matrix $\mathbf{G}$. The sum of these two values is the incident vector of category sizes with $d$ incidents at time point $t_n$. The bottom part of Equation (18) computes the incidence row vector at time point $t_n$ as the incidence row vector at time point $t_n$ times $\mathbf{G}$. The sum of the entries in row vector $\mathbf{D}(n)_d$ is the size of the segment at time point $n$ for $d$ incidents in Figure 2d.

$$\mathbf{D}(n)_d = \left\langle \begin{array}{l} \text{if } 0 = d, \text{ then } \mathbf{D}(n)_0 = \mathbf{D}(n-1)_0\mathbf{P} \text{ where } \mathbf{D}(0)_0 = \mathbf{S}(0) \\ \text{if } 0 < d < n, \text{ then } \mathbf{D}(n)_d = \mathbf{D}(n-1)_d\mathbf{P} + \mathbf{D}(n-1)_{d-1}\mathbf{G} \\ \text{if } 0 < d = n, \text{ then } \mathbf{D}(n)_d = \mathbf{D}(n-1)_{d-1}\mathbf{G} \end{array} \right. \tag{18}$$

Figure 2f derives from Equation (19). Equation (19) gives the row vector $\mathbf{S}(n)$ expressed as the product of the row vector $\mathbf{S}(0)$ and the Markov matrix $\mathbf{M}$ raised to the power $n$. We define matrix $\mathbf{R}(n)$ and matrix $\mathbf{H}(n)$ such that their sum equals $\mathbf{M}^n$ similar to how Equation (16) expresses matrix $\mathbf{M}$ as the sum of $\mathbf{P}$ and $\mathbf{G}$. Specifically, the diagonal entries of $\mathbf{R}(n)$ are the same as diagonal entries of $\mathbf{M}^n$ while the off-diagonal entries of $\mathbf{R}(0)$ are zero. Therefore, $\mathbf{S}(0)\mathbf{R}(n)$ is a row vector of $J$ entries that give the sizes of the portion of the extent that is category $j$ at both time 0 and time $n$. These $J$ sizes in $\mathbf{S}(0)\mathbf{R}(n)$ appear in Figure 2f as Persistence or Return for the categories. The diagonal entries of $\mathbf{H}(n)$ zero and the off-diagonal entries of $\mathbf{H}(0)$ are the same as the off-diagonal entries of $\mathbf{M}^n$. Therefore, $\mathbf{S}(0)\mathbf{B}(n)$ is a row vector of $J$ entries that give the size of the portion of the extent that is category $j$ at time $n$ and not $j$ at time 0. Those $J$ sizes in $\mathbf{S}(0)\mathbf{B}(n)$ appear in Figure 2f as the Gain categories.

$$\mathbf{S}(n) = \mathbf{S}(0)\mathbf{M}^n = \mathbf{S}(0)\mathbf{R}(n) + \mathbf{S}(0)\mathbf{B}(n) \tag{19}$$

*2.3. Case Studies*

We use two case studies to demonstrate and compare the characteristics of the Flow and Markov matrices. Both case studies derive from the Plum Island Ecosystems site of the Long Term Ecological Research network of the United States National Science Foundation. The first case study uses reference data from 1938, 1971, and 2013 for three land categories in a wetland landscape. The second case study uses reference data from 1971, 1985, and 1999 for three land categories in a suburban landscape. The two earlier time points form the calibration time interval, from which we extrapolate using the Flow and Markov methods. We then compare the extrapolated changes to reference changes. Pattern validation assesses each extrapolation's predictive power by comparing the extrapolated change to the reference change.

## 3. Results

*3.1. Wetland Case Study*

Figure 3 shows the first case study's data, which derive from maps digitized by a team from the United States Long Term Ecological Research network [25]. Data availability dictated the spatial extent. The 10 meter resolution raster map shows the cumulative incidents of land change during two time intervals for the categories Marsh, Water, and Other. The gray shades indicate persistence during both the calibration time interval from 1938 to 1971 and the subsequent time interval from 1971 to 2013. Non-gray indicates the category at 2013. Lighter shades of blue for Water, green for Marsh, and yellow for Other indicate a single transition either from 1938 to 1971 or from 1971 to 2013. Darker shades indicate transition during both time intervals. Changes are concentrated along the coast, meaning between places that are always Water and places that are always Marsh.

The bar chart in Figure 3 shows the size of the spatial extent that is different from 1938 for each of the four bars. The first bar on the left is the change during the calibration interval from 1938 to 1971. The second and third bars show the same 33-year calibration interval combined with a 33-year extrapolation interval. The second bar extrapolates change via the Flow extrapolation. The Flow extrapolation consists entirely of one incident during 66 years from 1938 to 2004. The size of change via the Flow matrix during the 66 years is twice the size of change during 33 years. The third bar demonstrates change via the Markov extrapolation, which is shorter than the Flow bar because of two reasons. The Markov extrapolation implies deceleration, and the Markov extrapolation assumes some part of the spatial extent changes more than once. The top of the bar shows segments with two incidents, indicating change during both the calibration and extrapolation intervals. The bottom of the bar shows segments with one incident, indicating change during either the calibration or the extrapolation interval. The fourth bar reveals the change according to the reference data during 75 years, meaning the 33-year interval from 1938 to 1971 combined with the 42-year interval from 1971 to 2013. The fourth bar is shorter than both the Flow and Markov bars, which indicates that the part of the extent that experiences at least one

change in the reference data is smaller than either extrapolation portrays. Furthermore, the portion of the change that derives from two incidents is greater in the reference data than in the extrapolations.

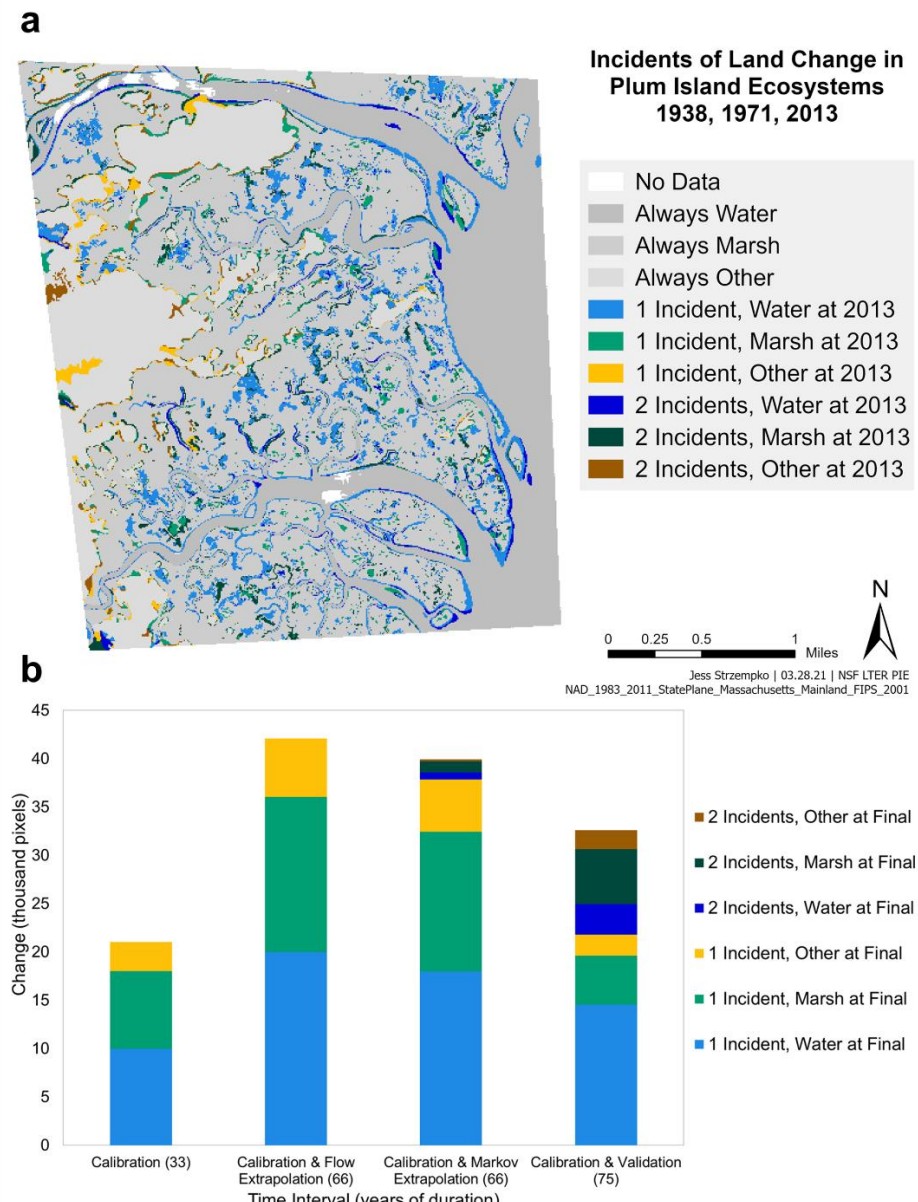

**Figure 3.** (**a**) Map and (**b**) bars for the cumulative number of incidents during two time intervals along with the category at the end of the second time interval for categories Marsh, Water, and Other. Final means 1971 for the Calibration (33) bar, 2004 for the two Extrapolation (66) bars, and 2013 for the Validation (75) bar.

Figure 4 compares the behavior of the Flow and Markov extrapolations similar to Figure 2. The left column of graphs displays the Flow extrapolation while the right column displays the Markov extrapolation. The graphs show the calibration interval's start year at 1938 and end year at 1971. Figure 4 facilitates comparison by showing both methods extrapolated to the year 2235. The Flow extrapolation cannot extrapolate beyond 2247 because Marsh reaches zero persistence at 2247. The Markov extrapolation occurs during discrete 33-year increments; thus, 2235 is the last year that is less than 2247. The Markov graphs show ten time points, which form the calibration time interval and eight extrapolated time intervals.

Figure 4a,b show the size of each category at each time point, with Water at the top, Marsh in the middle, and Other at the bottom. Both Flow and Markov extrapolate an increase in Water's size and a decrease in Marsh's size. The size of each category is a linear function of time in the Flow extrapolation. The size of each category asymptotically approaches a constant size as time progresses in the Markov extrapolation.

Figure 4c,d demonstrate the accumulation of incidents for each method. The Flow extrapolation has either zero or one incident per observation. The Markov matrix allows an additional incident for each observation during each additional time interval. Some portions of the extent accumulate up to nine incidents from 1938 to 2235.

Figure 4e,f show change since 1938 at the bottom of each graph. Figure 4a,e contain identical segments in a different order from top to bottom for the Flow method. Return of a category is the size of the extent that began at 1938 as a particular category, transitioned from that category during a subsequent extrapolation interval, and then returned to the initial category by the end time. The Flow matrix has at most one incident; thus, the Flow matrix does not have Return.

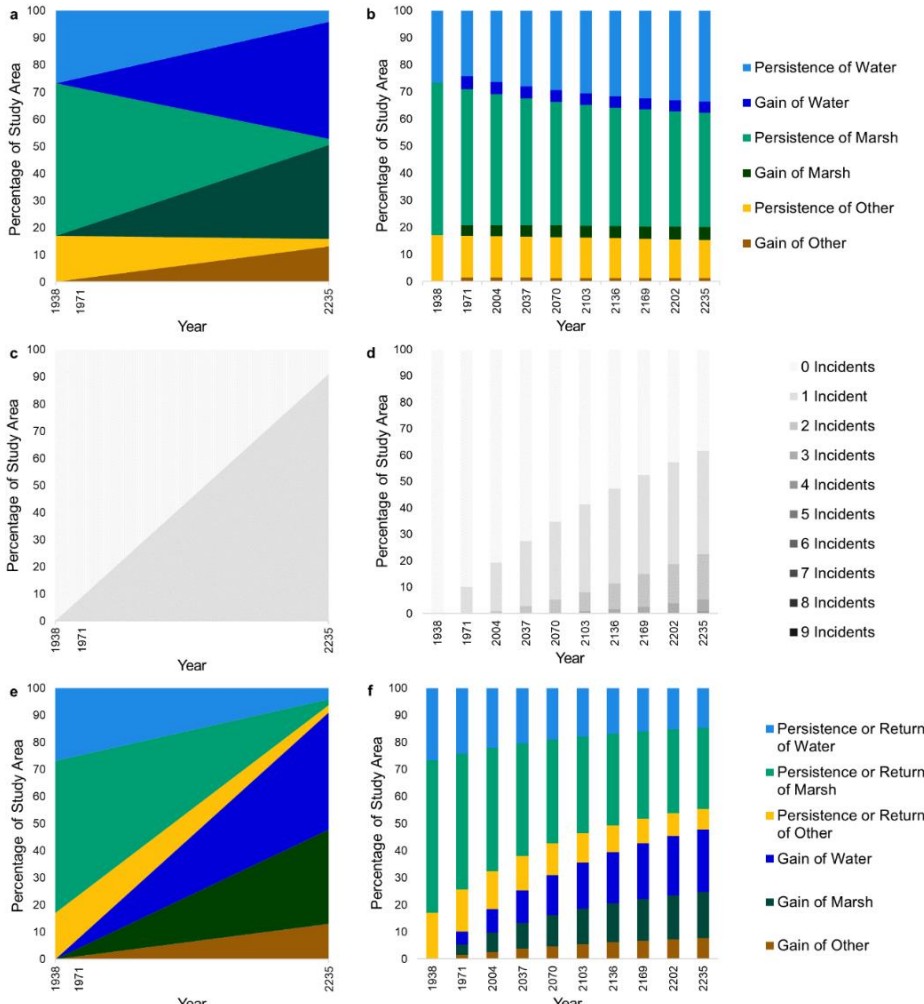

**Figure 4.** The three graphs on the left show the Flow extrapolation while the three graphs on the right show the Markov extrapolation from a calibration interval that starts at 1938 and ends at 1971. One legend applies to each pair of graphs. The upper pair shows the size of each category via (**a**) Flow and (**b**) Markov, where each Markov bar shows transitions from the previous 14 years. The middle pair shows the cumulative number of incidents via (**c**) Flow and (**d**) Markov. The lower pair shows differences from 1938 in the bottom three segments via (**e**) Flow and (**f**) Markov extrapolations.

### *3.2. Suburban Case Study*

The second case study uses data from the state of Massachusetts [26], which describe 21 land-use categories that we merged into three categories to illustrate the concepts. The raster maps have 30 meter resolution pixels at three time points for three Massachusetts suburban towns: Topsfield, Hamilton, and Wenham. Figure 5 is a map of the cumulative incidents of land change for three categories: Built, Forest, and Other. The gray color in the map indicates persistence during the calibration time interval from 1971 to 1985 and the subsequent time interval from 1985 to 1999. The categories at 1999 indicate transitions to Built in blue, Forest in green, and Other in yellow. Lighter shades indicate a single transition during either 1971–1985 or 1985–1999. Darker shades indicate transitions during both time intervals. Changes are primarily to Built.

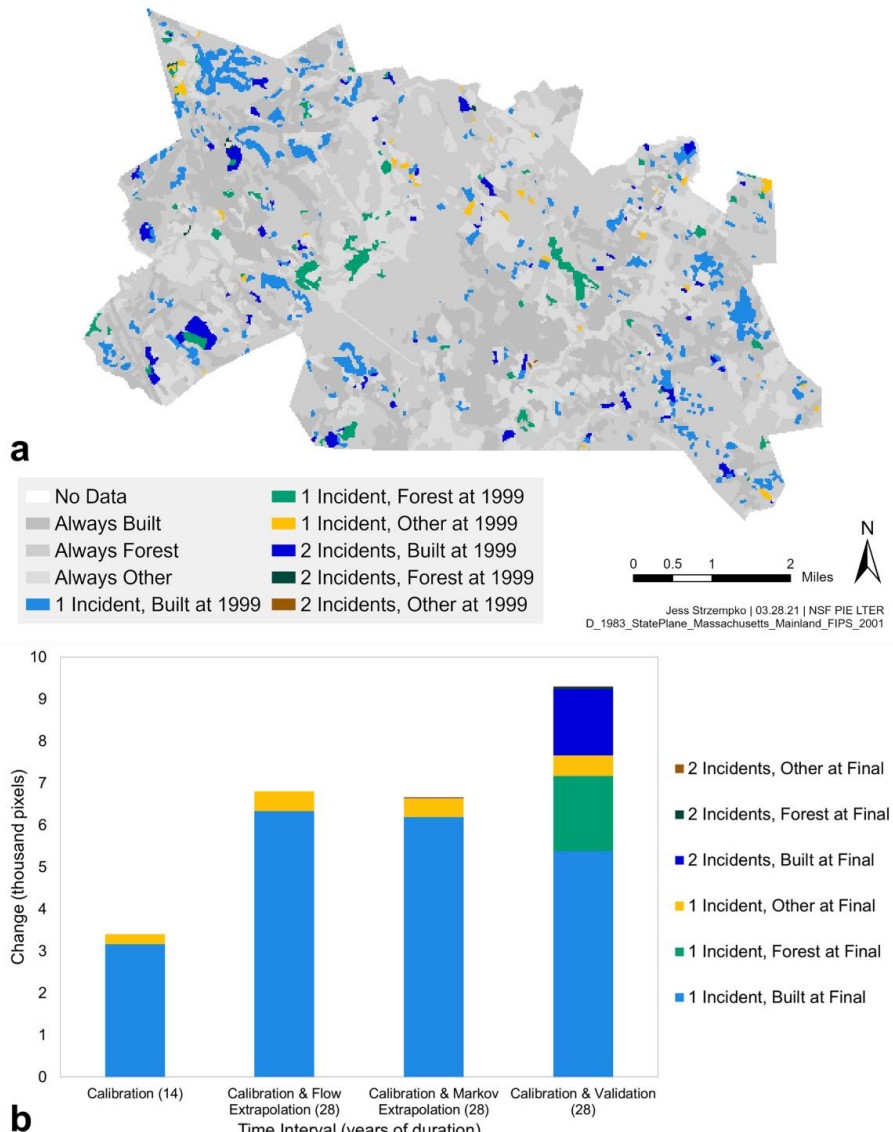

**Figure 5.** (**a**) Map and (**b**) bars for the cumulative number of incidents during two time intervals along with the category at the end of the second time interval for categories Built, Forest, and Other. Final means 1985 for the first Calibration (14) bar whereas Final means 1999 for the other three bars.

The bar chart in Figure 5 shows cumulative incidents for four bars. The vertical axis is the size that experiences at least one change. The first bar on the left is the change during the calibration interval from 1971 to 1985, during which the predominant transition is to Built. The second bar combines the 14-year calibration interval with the 14-year Flow

extrapolation. The second bar is twice the size of the first bar. The third bar combines the 14-year calibration interval with the subsequent 14-year Markov extrapolation thus shows one or two incidents. The area of change from 1971 for the Markov extrapolation is slightly smaller than for the Markov extrapolation because of two reasons. Markov extrapolates deceleration, and Markov assumes some of the spatial extent changes more than once. The fourth bar at the right combines the calibration interval and the subsequent interval from 1985 to 1999 according to the data in the map. The fourth bar is nearly three times the size of the first bar because the reference data show the change accelerates, which is a phenomenon that neither of the extrapolation methods can portray. The Flow and Markov methods extrapolate Built's gain but not Forest's gain because Forest did not gain during the calibration interval.

Figure 6 compares the behavior of the Flow and Markov extrapolations, similar to Figures 2 and 4. As before, the left column of graphs displays the Flow extrapolation while the right column of graphs displays the Markov extrapolation. We extrapolated to 2097 with both the Flow and Markov for direct comparison. We chose an endpoint of nine time intervals of 14 years each to match the number of time intervals of the previous case study. The Flow extrapolation cannot extend beyond when a category reaches zero persistence, which forest does at 2262.

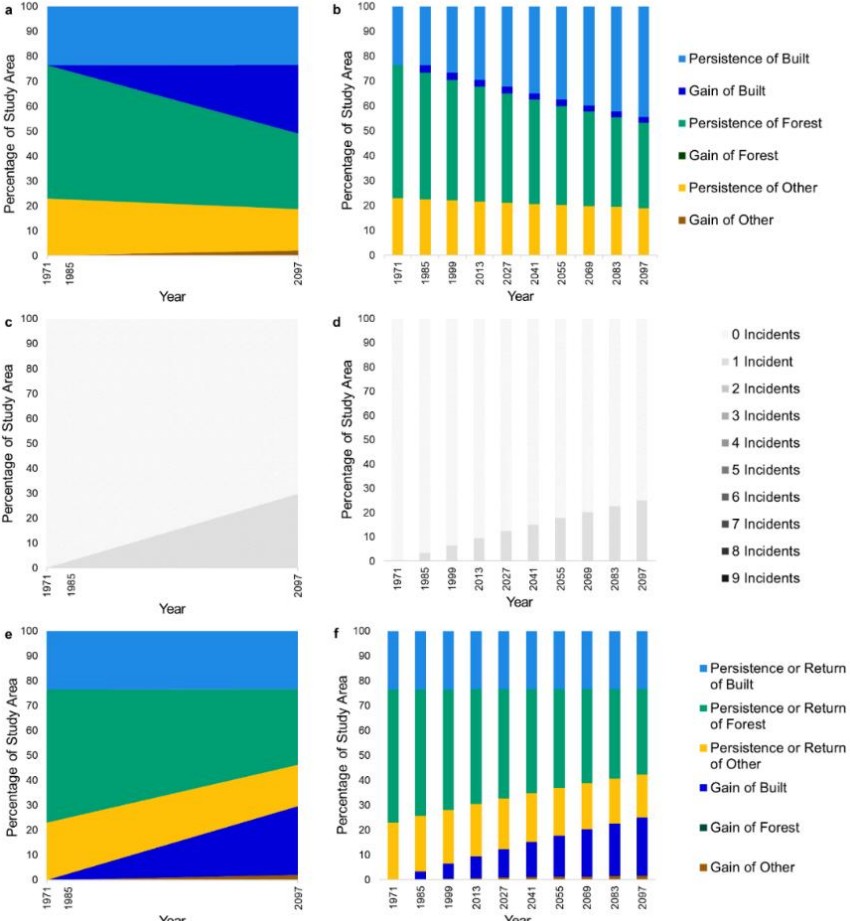

**Figure 6.** The three graphs on the left show the Flow extrapolation while the three graphs on the right show the Markov extrapolation from a calibration interval that starts at 1971 and ends at 1985. One legend applies to each pair of graphs. The upper pair shows the size of each category via (**a**) Flow and (**b**) Markov, where each Markov bar shows transitions from the previous 14 years. The middle pair shows the cumulative number of incidents via (**c**) Flow and (**d**) Markov. The lower pair shows the difference from 1971 in the bottom three segments via (**e**) Flow and (**f**) Markov extrapolations.

Figure 6a,b present the size of categorical persistence and gain as a function of time. Persistence and gain are from 1971 for the Flow extrapolation and from the previous time point for the Markov extrapolation. The Flow method extrapolates continuously from 1971 to 2097, while the Markov method extrapolates during nine discrete time intervals of 14 years. The light blue segment of Built persistence is constant because Built does not lose during the calibration interval. There is no Forest gain during the extrapolation because Forest does not gain during the calibration interval.

Figure 6c,d demonstrate the accumulation of incidents for each extrapolation method. The Flow extrapolation allows a maximum of one incident. The Flow extrapolation shows 30% of the study area changes by 2097. Incidents accumulate during each time interval for the Markov extrapolation; thus, some of the study area has nine incidents in 2097. The Markov extrapolation shows 25% of the study area changes at least once by 2097.

Figure 6e,f present each category's derivation from 1971. Gain means an increase when compared with 1971. Persistence means no change since 1971. Return means at least one loss and then subsequent gain since 1971. Figure 6f displays the discrete change of the Markov method where category sizes asymptotically approach constants.

## 4. Discussion

### 4.1. Comparison of Characteristics

The Flow and Markov matrices offer contrasting paradigms to describe and extrapolate transitions among categories. Table 2 summarizes the characteristics of the Flow and Markov extrapolation methods. The subsequent paragraphs elaborate on the characteristics in Table 2 first for the Flow extrapolation and then for the Markov extrapolation.

**Table 2.** Characteristics of Flow and Markov extrapolations.

| Characteristic | Flow | Markov |
|---|---|---|
| Necessary mathematical knowledge | Line | Matrix Algebra |
| Extrapolates at most one incident of change | Yes | No |
| Extrapolates through continuous time | Yes | No |
| Assumes constant size change per time change | Yes | No |
| Computes time point to reach zero persistence | Yes | No |
| Extrapolates to any desired time point | Constrained | Maybe |
| Extrapolates into the infinite future | No | Yes |
| Category sizes approach a steady state | No | Frequently |
| Category's gain depends on sizes of other categories | No | Yes |
| Can extrapolate acceleration of change | No | No |
| Can extrapolate transitions that calibration lacks | No | Yes |
| Matches true systems through a time series | Testable | Testable |

A fundamental difference between the Flow and Markov extrapolations is the mathematical knowledge required to interpret their behaviors properly. The Flow extrapolation requires the modeler to understand the equation of a line. The Flow matrix extrapolates at most one incident of change for each observation, which allows the Flow matrix to extrapolate through continuous time. If an extrapolation allows more than one incident of change, then the extrapolation must designate distinct intermediate time points when observations renew their candidacy for additional incidents of change. The Flow method extrapolates a constant change of size per change of time, which is equal to the calibration interval's change of size per change of time. This implies a constant decrease of persistence size per change of time, which allows the computation of the time point when a category reaches zero persistence from the start of the calibration interval. Consequently, the Flow extrapolation extends to any time point on the continuum from the end of the calibration interval to the time point when a category reaches zero persistence. The Flow extrapolation is constrained to not extrapolate the linear pattern beyond the first time point when a category reaches zero persistence. If change exists during the calibration time interval, then the Flow extrapolation does not extrapolate infinitely in the future, in which case the category sizes do not approach a steady state.

The Flow extrapolation's annual size of gain to category *j* is independent of the sizes of the categories that are not *j* from time $t_1$ to $T$. This can be a helpful characteristic when the sizes of the other categories are arbitrary, as is the case in Figure 3 for the Plum Island Ecosystems (PIE) where the size of water is arbitrary. The spatial extent in PIE derives from the bounds of a remotely sensed image, which contains an arbitrary size of water. The size of water's persistence in the eastern part of PIE does not influence the sizes of the annual transitions during the Flow extrapolation. However, the inclusion of more persistence in the spatial extent would extend the duration till a category reaches zero persistence.

The Flow extrapolation portrays a constant change in size per change in time and thus portrays neither acceleration nor deceleration of change. This makes sense because the calibration time interval has exactly two time points and thus cannot give evidence for acceleration or deceleration of change. Furthermore, exactly two time points cannot give evidence for more than one incident of change per observation. In these respects, the Flow extrapolation does not assume a process of change that is more complicated than the two time points of the calibration interval reveal. Consequently, if a transition from category *i* to category *j* has zero size during the calibration interval, then the transition has zero size during the Flow extrapolation.

In contrast to the linear Flow extrapolation, the Markov extrapolation requires modelers to understand the more complicated behavior of matrix algebra, which can be daunting. Matrix multiplication generates a Markov chain that assumes a process for which a calibration time interval between exactly two points cannot give complete evidence. The Markov extrapolation allows more than one incident of change, while the calibration interval cannot reveal whether an observation experienced more than one incident of change in the system. The Markov method extrapolates iteratively during *n* discrete time intervals allowing zero or one incident per iteration. Thus, the Markov extrapolation can generate portions of the extent that accumulate *n* incidents by time $t_n$. An extrapolation that allows observations to have more than one incident of change must have time points at which the observations renew their candidacy for change. The Markov extrapolation allows an additional incident of change by proceeding through time increments as the distinct bars of Figures 2, 4 and 6 show. Each additional time increment allows an additional incident of change. The Markov matrix does not extrapolate through continuous time. The Markov extrapolation computes the size of change during each time increment by multiplying a constant proportion of each losing category times the size of the losing category at the start of the increment. This frequently leads to a deceleration of the size of change, not a constant size change per time change. If a category's diagonal entry in the Markov matrix is neither zero nor one, then persistence from the start of the calibration interval shrinks exponentially but does not reach zero, so the Markov extrapolation cannot compute a time point when persistence reaches zero. This allows the Markov extrapolation to extend into the infinite future via incremental time steps. The Markov extrapolation frequently implies that change decelerates to zero as the sizes of the categories approach a steady state dictated by a limiting distribution.

The size of Markov's extrapolated gain to category *j* can be influenced by the sizes of the categories that are not *j*. This creates a complication when the size of some of the categories are arbitrary, such as the size of water in the PIE example. The arbitrary size of water's persistence in PIE influences the extrapolated sizes of all the transitions in PIE. This characteristic is especially problematic in the popular application to urban growth simulation models where non-urban transitions to urban while urban persists. The shrinkage of non-urban dictates the gain of urban during the Markov extrapolation, but the size of non-urban is frequently arbitrary.

The calibration interval has exactly two time points. Two time points cannot give evidence of whether change is accelerating, decelerating, or neither. Nevertheless, Markov extrapolations tend to compute decelerating change and cannot portray accelerating change. Researchers frequently select a study site because change is accelerating in a manner that cannot continue indefinitely, in which case it makes little sense to use a Markov extrapolation that shows decelerating change that approaches a steady state in the infinite

future. The Markov extrapolation assumes a process of change that is more complicated than the two time points of the calibration interval can reveal. For example, consider three land categories: Forest, Agriculture, and Urban. The calibration interval over a short duration might show zero transition from Forest to Urban but positive transitions from Forest to Agriculture and from Agriculture to Urban. The Markov extrapolation across two time intervals would show a transition from Forest to Urban. In this manner, the Markov extrapolation across multiple time intervals can show transitions that do not exist during the shorter calibration time interval.

### 4.2. Extrapolation to Specific Time Points

Discrete incremental extrapolation is necessary to allow more than one incident. The Markov matrix allows more than one incident by extrapolating incrementally using discrete time intervals that are equal in duration to the calibration time interval. This can cause a mathematical challenge when a modeler wants to extrapolate for a duration that is not a whole-number multiple of the duration of the calibration interval. For example, consider a calibration interval from the year 2000 to the year 2008, where the modeler wants to extrapolate from 2008 to 2010 and 2020. The modeler wants to extrapolate beyond the calibration time interval across durations that are not multiples of the duration of the calibration interval. The modeler must convert the 8-year Markov matrix to a Matrix that extrapolates during intervals of both 2 years and 12 years. It would be helpful to have a method to convert the 8-year Markov matrix to an equivalent Markov matrix that portrays an arbitrary duration, such as 1, 2, or 12 years. However, the procedure to convert a non-annual calibrated Markov matrix to an equivalent annual Markov matrix might lack a unique solution where the entries in the annual Matrix have real values on the inclusive interval from zero to one. For some matrices, the only solutions have negative numbers or numbers that involve the square root of negative one, which defies practical interpretation [22,23]. For other matrices, multiple solutions exist.

Equation (20) gives an example where **M** is a 2-by-2 Markov matrix that derives from a calibration time interval that has an even number of years, during which 0.18 of Category 1 transitions to Category 2 while 0.18 of Category 2 transitions to Category 1. Two solutions exist when computing a Markov matrix for half of the duration of the calibration time interval. Equation (20) shows that one solution is when 0.1 of Category 1 transitions to Category 2 while 0.1 of Category 2 transitions to Category 1. The matrix on the extreme right in Equation (20) shows that a second solution is where 0.9 of Category 1 transitions to Category 2 while 0.9 of Category 2 transitions to Category 1. If $m_{11} + m_{22} > 1$, then no real-numbered solutions exist to compute a Markov matrix that has a duration half of the calibration interval's duration.

$$\mathbf{M} = \begin{bmatrix} m_{11} & m_{12} \\ m_{21} & m_{22} \end{bmatrix} = \begin{bmatrix} .82 & .18 \\ .18 & .82 \end{bmatrix} = \begin{bmatrix} .9 & .1 \\ .1 & .9 \end{bmatrix}^2 = \begin{bmatrix} .1 & .9 \\ .9 & .1 \end{bmatrix}^2 \tag{20}$$

Equation (21) shows a case for categories that cycle with each other, such as in shifting cultivation. The two time points that bound the calibration interval might show no change, in which case the calibration matrix is the Identity matrix **I**. If the calibration interval is 6 years, then one solution for an annual Markov matrix is **I** because **I** raised to any whole numbered power is **I**. However, several solutions exist that show change during annual time steps. Equation (21) gives five solutions for an annual matrix when raised to the power 6 produces the Identity matrix **I**. The two time points that bound the calibration interval lack information to determine a single solution.

$$\mathbf{I} = \begin{bmatrix} 0 & 1 & 0 \\ 0 & 0 & 1 \\ 1 & 0 & 0 \end{bmatrix}^6 = \begin{bmatrix} 0 & 0 & 1 \\ 1 & 0 & 0 \\ 0 & 1 & 0 \end{bmatrix}^6 = \begin{bmatrix} 0 & 1 & 0 \\ 1 & 0 & 0 \\ 0 & 0 & 1 \end{bmatrix}^6 = \begin{bmatrix} 1 & 0 & 0 \\ 0 & 0 & 1 \\ 0 & 1 & 0 \end{bmatrix}^6 = \begin{bmatrix} 0 & 0 & 1 \\ 0 & 1 & 0 \\ 1 & 0 & 0 \end{bmatrix}^6 \tag{21}$$

The Land Change Modeler in the TerrSet software uses a regression-based approximation to extrapolate to any time point, even when the time point does not derive from a whole number multiple of the duration of the calibration interval [12]. The regression-based method quickly produces a unique solution that is plausible for applications to land change modeling. The DINAMICA software can also compute a Markov matrix for a time interval that does not equal the calibration time interval [7]. Furthermore, both DINAMICA and TerrSet's Land Change Modeler allow modelers to enter a customized matrix that portrays change to any particular time point. Therefore, modelers may enter the Flow extrapolation into the software, which will then distribute the Flow transitions spatially. We used Excel to compute the results in our manuscript because pre-programmed software packages do not yet include the Flow matrix. We encourage readers to use this manuscript's equations to write code for the Flow matrix.

*4.3. Which Method to Choose?*

Modelers will naturally wonder whether to use the Flow, Markov, or some other extrapolation method. The answer depends on the purpose of the modeling. If the purpose is to communicate to a non-technical audience, then the Flow extrapolation has the advantage of being easier to explain and understand. If the purpose is to extrapolate more than one change during sequential time intervals of the extrapolation, then the Flow matrix is insufficient while the Markov matrix is an option. However, the simulation modeling applications that we know simulate at most one change in a pixel. If the purpose is to extrapolate the pattern of change, then the modeler should examine the pattern of change through multiple time intervals to find the extrapolation that is a better fit. Several methods examine the patterns of change across sequential time intervals [27,28]. For example, transition-level Intensity Analysis compares the Markov matrix for each time interval to see whether the transition intensities are stationary across time intervals [29,30].

If empirical analysis reveals that a Markov process has been operating across several time intervals, then it might make sense to apply a Markov process to extrapolate change. However, if a Markov process has been operating across several time intervals, then the sizes of the categories may have already approached a steady state, in which case extrapolation will not produce substantial additional changes.

If the purpose is to simulate the hypothesized process of change, then the modeler must match the mathematical behavior of the extrapolation with the hypothesized process of change. We have no reason to assume that natural or human-managed landscapes follow the Flow matrix or the Markov matrix. If modelers lack a hypothesized process of change, then modelers can use an empirically based pattern of change. The Flow and Markov matrices offer distinct ways to describe the patterns during the calibration interval. Modelers could apply both the Flow and Markov extrapolations to see whether the difference between them has practical importance.

## 5. Conclusions

The Flow matrix is a novel and straightforward method to describe and extrapolate transitions among categories. The Flow matrix avoids the mathematical problems of the popular Markov matrix. Our case studies demonstrate the differences between the Flow and Markov matrices concerning the temporal characteristics of extrapolation, the temporal extent of the extrapolation, the sensitivity to persistence during the calibration interval, and the allowance of a category's start size to equal zero. The Flow matrix expands the possibilities for how a modeler can envision system dynamics. Each matrix has distinct characteristics that might be advantages, disadvantages, or neither, depending on the modeler's goals. Modelers should consider the trade-offs of each approach when extrapolating transitions among categories.

**Author Contributions:** Conceptualization, R.G.P.J.; Formal analysis, J.S.; Funding acquisition, R.G.P.J.; Software, J.S.; Writing—original draft, J.S.; Writing—review and editing, R.G.P.J. All authors have read and agreed to the published version of the manuscript.

**Funding:** The United States National Science Foundation supported this work through its Long Term Ecological Research network via grants OCE-1637630 and OCE-2224608 to the Plum Island Ecosystems site. The Edna Bailey Sussman Foundation provided additional funding via grant 24608.

**Data Availability Statement:** This manuscript uses data that derive from the following links https://gce-lter.marsci.uga.edu/public/app/dataset_details.asp?accession=GIS-GCET-1810, accessed on 1 September 2023 and https://www.mass.gov/info-details/massgis-data-land-use-1951-1999, accessed on 1 September 2023.

**Acknowledgments:** Clark Labs facilitated this work by creating the GIS software TerrSet®.

**Conflicts of Interest:** The authors declare no conflict of interest.

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
