# Peer review of "The Flow Matrix Offers a Straightforward Alternative to the Problematic Markov Matrix"

_land, doi:10.3390/land12071471_

Round 1

Reviewer 1 Report

The manuscript offers a novel method entitled the flow matrix to extrapolate linearly the transitions from land use land cover maps as a novel alternative to Markov matrices. The approach is innovative for land change science. The Flow matrix as a novel methodological approach to extrapolate land cover is presented in an unusual way explaining the method using two examples illustrated in Figure 1 and Figure 2. Then the authors introduce formalism into the method in section 2.2.

The science in the manuscript sounds and some concerns have to be answered in the text to be ready for publication. My reviews suggest major revisions despite an unusual structure to introduce a novel method. The replicability of the method could be compromised in the future due to the unusual way of presenting the method. 

Specific comments:

Line 20: clarify the characteristics or give a hint of their nature. This phrase sounds airy.

Despite that the introduction section is short and concise, an extended background of the use of Markov matrices is needed to evidence cases in which the flow matrices can overperform in categories extrapolation processes.

In the last paragraph of the introduction section lines [46-47], the authors mention the abilities and disabilities of the flow matrix approach but in the rest of the manuscript, there is no evidence of the abilities or disabilities.

From the reader's point of view, in lines [30-32] Markov matrix method is explained and lines [42-44] offer similar meaning but from the flow matrix. To posse evidence of the alternativeness of the Flow matrix method over the Markov matrices, some contrasting arguments are needed here. 

I suggest to the authors first introduce the flow matrix instead of starting by using the illustrative example since there is no previous evidence or literature on the flow matrix method to understand the example. This could cause misunderstanding to the readers of Land Journal. Replicability may be implicated because of intricate methods section structure. This is why you must mention in line 80 cross-reference to section 2.2. Besides, section 2.2 is titled Illustrative Example and then you provide two examples. The first one with t0 and t1 in Figure 1 and then in Figure 2 you use a time interval of 5 years in extrapolation, please check this is not cleared up in the text.

Probably a comparative workflow between Markov matrices and the flow matrix in the methods section can improve understanding of the method since the example include values that are not formally introduced in the new method.

Line 55: authors mention t0 and t1 but these characteristics are not shown in Figure 1.

The use of category values in Figure 1 and then the use of t0 and t1 without units maybe is confusing, despite you explain the method in lines [58-59].

Line 67: what is the intention of using losses in Figure 1 (b) if Figure 2 is showed another example and losses are not used at all? This may be confusing the readers.

Once that formalism for the Flow matrix is presented there is no information about the tool to perform the analysis, if this was created in R, GIS (map algebra), TerrSet, etc. To ensure the replicability of the method you have to present how you perform the analysis, this means the tool(s) to get the results. The only hint appears in lines [493-496] in the discussion section but not in the methods.

In section 3.1 wetland case study, maps from 1938, 1971 and 2013 were used but there is no information or references about pixel resolution and how a map from 1938 which is not for sure a digital map was included in the analysis. This is because Figure 3 is presented the change in thousand of pixels meaning that the map was rasterized. If this was performed in other studies please add a reference. The resolution of the pixel seems fine since the graphical scale indicates 1 mile. Another aspect is if the time span in the first interval of 33 years is influencing the incidents compared to the second time interval of 42 years.

Line 229. In this line, the acronym Plum Islan Ecosystem (PIE) needs to be defined since this was also used in discussion section lines 391, 392 and 394.

Lines [240-247]: in this paragraph is not clear the intention of using the two case studies. Besides, the map categories were used as the original ones or did you merge categories to get three and facilitate calculations and interpretation of the Flow matrix and Markov matrices?

In Figure 3 I suggest indicating the upper part with (a) and the bar chart with (b). Greyscales were difficult to identify the 3 persistences. I suggest using some other contrasting colours or intensifying grey contrasts.

It's not clear to me the intention to perform 9 cycles of extrapolation until the year 2235. Despite that in lines 276-277 is mentioned the reason that the marsh reaches zero persistence. This can be true for categories such as forest to some extent that reaches zero persistence after a total shrinkage in the landscape. This can be a limit of the Flow matrix method.

Similar to the first case study, the second one needs more details about the maps used, pixel resolution and the tool to perform the spatial analysis of persistence, transitions, etc.

The description of Table 2 refers to a valuation of extrapolations with flow and Markov matrices. This is not only a characteristics list. Please check. This may be part of the abilities/disabilities of the Flow matrix.

Lines [372-405] A paragraph with 33 lines is too hard to understand. I suggest to the authors split it to increase readability.

Lines [406-450] A paragraph with 44 lines is too hard to understand. I suggest to the authors split it to increase readability.

A limitation of the Flow matrix is the constant decrease of persistence, lines[380-381]. This has to be highlighted in some parts of the text when using the extrapolation of categories intended for conservation as forests, mangroves, etc. 

You mention in line 392 about the extent of the remote sensing images but there is no data about this before. Even about an image for 1938, please explain.

In section 4.3 there is a sequence of modelling purposes and recommendations for multiple cases of categories extrapolations. However, those were not developed and treated in the manuscript. The interpretation of the reader seems that this was only discussed but not exemplified and assess quantitatively in this manuscript. Probably, examining some cases and adding this in supplementary material can support the discussion points in this section.

Reviewer 2 Report

The manuscript, entitled "The Flow matrix offers a simple alternative to the troublesome Markov matrix," addressed a very important and challenging issue in simulation modeling and how it could help develop modeling methods that make it easy for users to explore transitions. I appreciate that the authors took the time and effort to research this issue. Scholars in the relevant field will hopefully have access to some useful modeling approaches that are better than the ones they already have. Even though I think the author did a great job of modeling, I have some questions about whether or not the Flow matrix is better than the Markov matrix, some of the answers may be incorporated in the Introduction/Discussion section with proper literature support to help readers understand and use of the Flow matrix.

Queries:

1.     How does the flow matrix take temporal aspects of the relationships into account? Can it detect time-varying flows and changes in flow patterns?

2.     Can the flow matrix accommodate various entities or states? Is it applicable to a variety of problem domains and settings?

3.     What methods are used to train or estimate the flow matrix? Does it require labelled data for learning, or can it utilise unlabeled data?

4.     What kind and quantity of data is required to construct the flow matrix? Does it require particular data formats or sources?

5.     How easily is the flow matrix interpretable? Can it offer insights into the data's underlying patterns and relationships? Does it permit simple interpretation of the flow dynamics?

6.     How does the computational complexity of working with the flow matrix scale with data size or entity count? Can it effectively manage large-scale problems?

7.     How can the performance of the flow matrix model be measured? How does it compare with the Markov matrix in terms of precision, prediction, and other pertinent metrics?

8.     How resilient is the flow matrix model against noise, missing data, and outliers? Does it have effective mechanisms for handling data imperfections?

9.     Can other models or techniques be integrated with the flow matrix? Exist any unique considerations or obstacles when combining it with other modelling methods?
